# The Concept behind the Suitability of Menstrual Blood-Derived Stem Cells for the Management of Vaginal Atrophy among BRCA Mutation Carriers after RRSO

**DOI:** 10.3390/ijms25021025

**Published:** 2024-01-14

**Authors:** Mariana Robalo Cordeiro, Bárbara Laranjeiro, Margarida Figueiredo-Dias

**Affiliations:** 1Gynecology University Clinic, Faculty of Medicine, University of Coimbra, 3000-548 Coimbra, Portugal; barbaralaranjeiro@gmail.com (B.L.); marg.fig.dias@gmail.com (M.F.-D.); 2Gynecology Department, Hospital University Centre of Coimbra, 3004-561 Coimbra, Portugal

**Keywords:** menstrual blood-derived stem cells, vaginal atrophy, regenerative medicine, BRCA mutation, risk-reducing surgery

## Abstract

Risk-reducing bilateral salpingo-oophorectomy (RRSO) is recommended for breast cancer gene 1 (BRCA1) and 2 (BRCA2) mutation carriers. A major consequence of RRSO is surgical menopause associated with severe menopausal symptoms, mostly genitourinary complaints. Due to the inherent breast cancer risk, estrogen-based therapies are generally avoided in these patients. So far, the non-hormonal approaches available are not efficient to successfully treat the disabling vaginal atrophy-related symptoms. In regenerative medicine, mesenchymal stem cells (MSC) are the most frequently used cell type due to their remarkable and regenerative characteristics. Therapies based on MSC have revealed positive outcomes regarding symptoms and signs associated with vaginal atrophy by promoting angiogenesis, vaginal restoration, and the proliferation of vaginal mucosa cells. Menstrual blood-derived stem cells (MenSC) are a novel source of MSC, with promising therapeutic potential directly linked to their high proliferative rates; low immunogenicity; non-invasive, easy, and periodic acquisition; and almost no associated ethical issues. In this review, we update the current knowledge and research regarding the potential value of previously preserved MenSC in the therapy of vaginal atrophy among BRCA mutation carriers subjected to RRSO.

## 1. Introduction

Risk-reducing bilateral salpingo-oophorectomy (RRSO) is a surgical procedure that includes bilateral removal of the ovaries and fallopian tubes and is often recommended for individuals who carry breast cancer gene 1 (BRCA1) and 2 (BRCA2) mutations, which are associated with hereditary breast and ovarian cancer syndrome (HBOC) [1,2]. The optimal timing of RRSO depends on several factors, including age, medical history, family planning goals, and overall physical and emotional well-being [1,2,3,4]. Nevertheless, it is generally recommended that RRSO should be considered between the ages of 35 and 40 or when family planning is complete [3,5]. It is extremely important for women with BRCA mutation to have detailed discussions with their healthcare providers, genetic counselors, and specialists to make informed decisions about RRSO. Furthermore, ongoing medical follow-up and support are essential to address both the physical and emotional issues of this procedure.

A major consequence of RSSO is surgical menopause, which is often associated with severe menopausal symptoms, mainly genitourinary complaints that are strongly related to the hypoestrogenic-induced vaginal atrophy [3,6,7]. Due to the inherent breast cancer risk predisposition among BRCA mutations carriers, estrogen-based therapies are generally avoided, even though this issue remains controversial [8,9]. The non-hormonal treatment approaches recommended to treat the disabling symptoms associated with vaginal atrophy in these patients are over-the-counter vaginal lubricants and moisturizers, as well as hyaluronic acid local injections, vaginal CO2 laser, and radiofrequency treatments [10]. Unfortunately, so far, none of these have revealed optimal success [10].

So far, in regenerative medicine, mesenchymal stem cells (MSC) are the most frequently used cell type with proven positive outcomes regarding symptoms and signs associated with vaginal atrophy by promoting angiogenesis, vaginal restoration, and the proliferation of vaginal mucosa cells [10,11,12]. More than a decade ago, a novel source of MSC from human menstrual fluid, named menstrual blood-derived stem cells (MenSC), was discovered [13]. Since then, researchers have focused their attention on studying MenSC, which are a unique stem cell population with high proliferative rates, low immunogenicity, periodic acquisition in a non-invasive manner, and with almost no ethical issues [13]. Hence, the therapeutic potential of these cells has been explored in various diseases with surprising in vivo results [13].

Our review aims to update the current knowledge and research regarding the potential value of menstrual blood-derived stem cells in the therapy of vaginal atrophy, namely, among BRCA mutation carriers subjected to RRSO.

### 1.1. Risk-Reducing Bilateral Salpingo-Oophorectomy in BRCA Mutation Carriers

Approximately 84% of hereditary breast cancers and more than 90% of hereditary ovarian cancers are caused by mutations in BRCA1 and BRCA2 genes [5,14]. As these genes encode for proteins involved in tumor suppression, carriers of BRCA1 or BRCA2 mutations have deficient DNA-repair mechanisms and cell cycle dysregulation. The DNA damage induces genomic instability and subsequently increases susceptibility to tumorigenesis, increasing ovarian and breast cancer risk and worsening overall survival compared to the BRCA mutation-free population [4,9]. Carriers of BRCA1 and BRCA2 pathogenic variants have lifetime risks of ovarian cancer and fallopian tube cancer of 40–60% and 15–30%, respectively. In addition, both BRCA1 and BRCA2 mutations predispose to a breast cancer lifetime risk of greater than 60% [5].

Ovarian cancer is the gynecological tumor with the highest mortality rate, mostly because of the asymptomatic growth of the tumor and the delayed onset of symptoms. Due to the absence of effective screening or early detection methods, ovarian cancer is often diagnosed in advanced stages with poor prognosis. Risk-reducing bilateral salpingo-oophorectomy is the most efficient prophylactic strategy proposed by international guidelines for ovarian and fallopian tube cancers’ prevention among BRCA mutation carriers. RRSO should include the bilateral removal of both ovaries and fallopian tubes and should be reserved for patients at a high risk of epithelial ovarian and fallopian tube cancer [4,15].

Factors such as the pathogenic variant type, the patient’s preference, and family history should be taken into consideration when determining the optimal timing of RRSO. Performing RRSO before the recommended age can have a negative impact on a woman’s health, including all premature detrimental menopausal consequences such as an increased risk of heart disease, osteoporosis, vasomotor symptoms, sexual dysfunction, and neurocognitive deterioration, which are particularly significant if women have contraindications to hormone replacement therapy use; thus, appropriate timing is critical [2,5]. For this reason, this procedure is recommended after the completion of childbearing or at 35–40 years for BRCA1 mutation carriers or at 40–55 years for BRCA2 mutation carriers.

Data from several studies indicate that RRSO is associated with a decrease in ovarian cancer (96%) and breast cancer (50%) occurrence and with a significant reduction in all-cause mortality rate, especially in BRCA1 mutation carriers [1,5]. Furthermore, in hereditary breast and ovarian cancer that are non-BRCA-related, RRSO is also advised, but the optimal timing for RRSO remains unclear and dependent on which BRCA gene is mutated [6]. However, recent research has shown contradictory data regarding breast cancer risk reduction after RRSO, leading researchers to question the benefit of this procedure in this context [16]. To improve the informed counseling for surgical prevention offered to BRCA carriers, Gaba et al. performed the most comprehensive systematic review and meta-analysis on determining the impact of this surgery on breast cancer risk, which included 14 publications. Their results found that RRSO is not associated with a significant reduction in the overall primary breast cancer (PBC) or contralateral breast cancer (CBC) risk in BRCA1 and BRCA2 carriers combined or in BRCA1 carriers alone. In addition, for either BRCA mutation type alone, RRSO is not associated with a reduction in CBC risk. Nevertheless, RRSO is associated with a significant reduction in PBC risk in BRCA2 carriers alone and improved breast cancer survival in BRCA1 and BRCA2 carriers previously diagnosed with breast cancer [16].

Interestingly, based on the hypothesis that breast tumors diagnosed post-RRSO might be smaller and less aggressive compared to ones diagnosed pre-RRSO, a recent study published by Stuursma et al. evaluated the histopathological features of breast cancers that develop before and after RRSO in BRCA1/2 germline pathogenic variant carriers. The results of their research showed that breast cancer diagnosed post-RRSO tends to be smaller in size compared to breast cancer diagnosed pre-RRSO [17]. However, no differences were observed concerning the incidence of in situ carcinoma, breast cancer subtype, hormone and HER2 receptor expression, or tumor grade [17]. These findings highlight the need for extensive research on this topic so that accurate and final conclusions can be transmitted to these patients.

Ovarian cancer screening is not effective and most women are diagnosed with advanced disease. Moreover, the 5-year mortality rate of 48.6% highlights a dramatic and urgent need to develop and implement prevention strategies for this tumor [2,5]. Hence, even though controversy remains regarding RRSO and breast cancer risk reduction, the previously described evidence supports the recommendation of RRSO in these patients [2,5].

### 1.2. Gynecologic Consequences of RRSO

A major adverse event of RRSO in premenopausal women is the abrupt onset of menopause and the increase in non-cancer-related morbidity, such as the decline in quality of life, sexual dysfunction, genitourinary syndrome of menopause, and an increased risk of osteoporosis, metabolic syndrome, cardiovascular diseases, and cognitive impairment.

The majority of women with BRCA mutation (74%) choose to undergo preventive surgery and report being very satisfied with their choice, manifesting lower cancer-related anxiety after surgery [18]. However, RRSO-induced menopause is associated with more severe menopausal complaints, sexual dysfunction, bone loss, and cardiovascular risk when compared with spontaneous menopause. Thus, healthy carriers undergoing RRSO at a young age should be informed of short- and long-term health consequences of premature menopause [3,6,7]. Effective physician–patient communication plays a fundamental role throughout the entire counseling process, so that an informed and conscious decision is made before RRSO, which will improve the post-surgery physiological, sexual, and psychosocial distress [19].

Sexual side effects are the most frequently referred concerns post-RRSO, and women are surprised by the real impact of RRSO on their quality of life. The most reported sexual symptom is vaginal dryness (more than 50%), followed by dyspareunia and reduced libido, which are considered the strongest predictors of discontent concerning the decision to undergo RRSO. It was also reported that premenopausal women undergoing risk-reducing surgery experience a greater decline in sexual function than post-menopausal women [20,21].

The pathophysiology behind the complaints associated with vulvovaginal atrophy are directly linked to the susceptibility generated by estrogen levels’ decrease on genital tissue. The vulva and the vagina widely express alpha- and beta-estrogen receptors throughout the reproductive age, but the latter are lost after menopausal onset, which results in anatomopathological changes in these tissues [10,22]. The advent of menopause leads to the removal of estrogen action in maintaining the normal vaginal epithelium thickness, the trophism of the smooth muscle layer, the adequate vascularization, and the density of nerve endings. In addition, the loss of upregulation on fibroblasts within the extracellular matrix is responsible for the decrease in elasticity and strength of these genital tissues. Other local changes related to hypoestrogenism occur, such as the thinning of the vaginal epithelium and the alkalinization of vaginal pH due to the decrease in glycogen levels in the vagina [22,23].

### 1.3. Treatment Options for Vaginal Atrophy after RRSO

Vulvovaginal atrophy is considered to be chronic and progressive and does not resolve spontaneously unless treatment is started [22,24,25]. The main goals of vulvovaginal atrophy treatment are the improvement of symptoms and the recovery of normal physiology of the urogenital tract.

Lifestyle modifications are generally considered the backbone of almost every treatment plan in medicine. Indeed, patients are advised to quit smoking, which reduces estrogen circulating levels, and to lose weight in the case of obesity, to improve blood flow to the genitourinary region. These changes are even advised prior to the RRSO intervention [10].

Hormone replacement therapy (HRT), consisting of estrogen alone or combined estrogen and progestogen, is used to decrease menopausal complaints and produces a significant reduction of vasomotor symptoms after RRSO [25]. Nevertheless, HRT is less effective in surgical menopause due to the sudden cessation of hormone production compared to spontaneous menopause [26,27].

Due to the high probability of having breast cancer in BRCA mutation carrier patients, the use of HRT is controversial. While short-term HRT seems to decrease post-menopausal complaints and does not seem to increase the risk of breast cancer in BRCA1/2 mutation carriers without a personal history of breast cancer, some recent studies have suggested that this may be true for women aged up to 45 years; however, beyond that, an increased risk is considered. Even though short-term HRT may be offered after RRSO, the limitations and risks associated with HRT should be clearly communicated to patients [7,8,28,29]. On the contrary, HRT should always be avoided in patient with a history of breast cancer [25,27].

Ospemifene is the only approved oral medication for the treatment of vulvovaginal dryness and moderate-to-severe dyspareunia. This molecule is a selective estrogen receptor modulator with an agonist effect on the vaginal epithelium. Even though it has revealed successful results, its prescription is not safe for carriers of the BRCA mutation after RRSO [25].

As opposed to systemic HRT, which has limitations related to breast cancer risk, local therapies, including low-dose intravaginal estrogens, may be considered to manage genitourinary symptoms of surgical menopause, including vulvovaginal dryness and dyspareunia, as well as urinary symptoms like urinary urgency, dysuria, or recurrent urinary tract infection.

Some studies have observed that the use of vaginal estrogen was associated with lower rates of female sexual dysfunction (up to 78%), including improved lubrication and orgasm scores, whereas systemic HRT had no effect. This suggests that for patients primarily struggling with female sexual dysfunction rather than vasomotor symptoms, vaginal estrogen may be a successful treatment strategy. Furthermore, patients may find the breast cancer risk profile of topical estrogen more acceptable in comparison to that of systemic combined HRT [20]. However, changes in serum estradiol levels induced by topical estrogens should not be overlooked. Data from a recent systematic review and meta-analysis on this issue concluded that even though low-dose vaginal estrogen shows the smallest serum estradiol levels’ changes, the safety related to cancer recurrence remains unclear in breast cancer survivors, especially for patients on aromatase inhibitors [30].

In patients with previous breast cancer, the use of topical estrogen is controversial and must be discussed case-by-case. Thus, the American Society of Clinical Oncology/American Cancer Society and the North American Menopause Society recommendations for first-line vulvovaginal atrophy treatment in this context are over-the-counter non-estrogenic vaginal lubricants and moisturizers. Lubricants may be water-, silicone-, or oil-based and should be applied to the external genitalia prior to sexual intercourse, to improve dyspareunia. On the other hand, vaginal moisturizers enhance genital hydration, helping to relieve the symptoms of vaginal dryness [25]. Other alternatives to hormone-based treatments for genitourinary symptoms include hyaluronic acid local injections and vaginal laser, even though their efficacy is limited, especially in cases of intense vulvovaginal atrophy. It has been noted that these local treatments do not reverse the progressive ageing of urogenital tissue [25].

A huge drawback related to topical treatment is therapeutic compliance, with only 45.2% of patients reporting satisfaction with local treatments [31]. The most common complaints reported for over-the-counter products (moisturizers and lubricants) are messiness and insufficient efficacy, while safety concerns were most frequently associated with estrogen-based therapies. The potential predisposition to develop cancer were the main concerns expressed by patients regarding the long-term use of topical hormonal treatment [31,32]. As a consequence of all these limitations, many women have discontinued topical treatment [31,32,33].

Novel non-hormonal approaches for vaginal atrophy must be researched, so that high-efficacy therapies can be offered to patients with a high risk of developing hormone-dependent cancers.

### 1.4. Menstrual Blood-Derived Stem Cells

The modern concept of stem cells define these unique undifferentiated population of cells by their clonality (usually arising from a single cell), their ability to reproduce themselves (self-renewal), to differentiate into different types of cells and tissue (potency), to persist for a long time, and to suffer regulation by the immediate environment (the niche) (Figure 1). Stem cells can be divided into two categories: the pluripotent growing in culture, also known as embryonic stem cells or induced pluripotent stem cells, and the tissue-specific stem cells [34,35]. Overall, they are considered to be responsible for the development, regeneration, and homeostasis of organ and tissue systems.

Menstrual blood-derived stem cells (MenSC) were first identified in 2007 and are referred to as a source of endometrial stem cells obtained from menstrual blood. MenSC primarily express CD9, CD13, CD29, CD41a, CD44, CD59, CD73, CD90, and CD105 but not CD19, CD34, CD45, CD117, CD130, or human leukocyte antigen-DR isotype (HLA-DR). In addition, there are some studies reporting a positive expression of embryonic and intracellular multipotent markers (OCT-4, c-kit proto-oncogene/CD117, and stage-specific embryonic antigen-4), which are absent in mesenchymal stem cells (MSC) from other sources [36].

MenSC have an extraordinary broad differentiation capacity into adipocytic, osteogenic, cardiomyocytic, and neuronal lineages, as well as respiratory epithelial, endothelial, myocytic, hepatic, germ-like, pancreatic cells, and ovarian tissue-like cells [13,36,37].

Recent studies have reported a very high proliferative rate among MenSC, being nearly twice as fast as bone marrow-MSC (BM-MSC), which can be partially explained by differences in telomerase activity between these cells [13,36,37]. This attribute contributes to the high expression of embryonic trophic factors and extracellular matrix in MenSC. Alongside the genomic stability observed in MenSC, the highly proliferating rate predicts unexpected therapeutic properties for these cells [36].

The immunomodulatory and anti-inflammatory capabilities of MenSC have mainly been described by in vitro studies through the interaction with a wide range of immune cells and the regulation of adaptive and innate immunity [13,36]. Via downregulation of the expression of cytokines GITR, GM-CSF, RANTES, MIP-1γ, eotaxin, MCP-5, and CCL1, MenSC could suppress MLE-12 cells apoptosis. In addition, these stem cells negatively influence PI3K/Akt/mTOR/IKK signaling mediated by TLR4, which disables the translocation of p-NF-κBp65 into the nucleus, resulting in a decrease in inflammatory cytokine. Moreover, Chen, L. et al. observed the MenSC secretion of interleukin−6 and interleukin−10 leading to the maturation of human blood monocyte-derived dendritic cells [13,36].

The paracrine effects of MenSC are strictly related to their secretion of important chemokines and cytokines, such as MMP-3, MMP-10, IL-4, hypoxia inducible factor-1 alpha, TGF-β1, TGF-β2, EGF, PDGF, nitric oxide, and insulin-like growth factor-1, which are crucial for tissue and organ regeneration [36].

MenSC are believed to also play an important role in angiogenesis, mostly through activation of the AKT and ERK pathways and expression of angiogenic factors such as VEGFR1, VEGFR2, eNOS, VEGFA, and TIE2 [36].

### 1.5. MenSC-Based Therapies Novel Approaches

The use of MenSC in cellular therapy remains unclear in clinical trials. Nevertheless, MenSC have already been extensively applied in preclinical studies and in some clinical research, with many of them revealing positive and successful outcomes in a variety of diseases. Gynecological applications of this novel therapy are summarized in Figure 2 [13,37,38,39,40,41,42,43,44]. Some novel strategies for the use of MenSC in treatment have been comprehensively studied.

CRISPR/Cas9 is a recently discovered genome editing technology already widely used in gene therapy. Efficient delivery of CRISPR/Cas9 components (guide RNA and Cas9 protein, which together form a ribonucleoprotein complex) to target cells is a critical consideration [45]. Various delivery methods, such as viral vectors or nanoparticles, are being explored to improve the efficiency and specificity of gene editing. Indeed, one of the challenges associated with CRISPR/Cas9 technology is the potential for off-target effects, where unintended genomic modifications may occur [46]. Deryabin et al. recently described that genetic manipulations of MenSC via CRISPR/Cas9 technology were successful in targeting plasminogen activator inhibitor-1 [37,39]. Hence, CRISPR/Cas9 can be used to boost MenSC capability to reprogram and differentiate in disease models through precise genetic manipulation [37,39].

Exosomes are defined as extracellular vesicles secreted by almost all cell types upon the fusion of multivesicular bodies and plasma membrane. These nanoscale-sized vesicles contain different types of cargo molecules, such as lipids, proteins, DNA, mRNA, and miRNA, which allows them to become novel intercellular communicators and vehicles for the delivery of therapeutical biomolecules to target cells [47]. There is evidence suggesting that exosomes secreted by MenSC could be considered a new type of cell-free treatment, with all the ethical and immune rejection issues benefits. In vitro results confirmed that MenSC-secreted exosomes mediate important cellular signaling pathways, including nuclear factor-κB (NF-κB) and PTEN/AKT/PKB pathways [37,39].

The stem cell niche is the cellular and molecular microenvironment where stem cells both reside and receive stimuli that shape their function and fate. Researchers are exploring the possibility of engineering artificial niches to control and enhance the behavior of transplanted stem cells [34]. This involves creating biomimetic environments that replicate key features of the native stem cell niche. On the other hand, strategies for delivering stem cells directly to the appropriate niche in the body can improve the engraftment and integration of transplanted cells into the host tissue [34]. Interestingly, age-related changes in the stem cell niche can affect the regenerative capacity of tissue. Hence, understanding and addressing these changes are extremely important for developing effective therapies, especially in older individuals [48]. The MenSC niche has not yet been accurately investigated, but mapping these niche cells will definitely promote the identification of future therapeutic targets [34,37].

The single-cell RNA sequencing (scRNA-seq) technology is a unique system to profile, identify, classify, and discover new or rare cell types from different organs and tissue and has become a powerful tool in biomedical research. Recently, it was observed that scRNA-seq can directly detect transcriptome information in single cells of MenSC [49]. In the future, the comprehensive and profound information given by this technology will allow it to develop drug models for precisely targeting specific genes or proteins to treat several diseases [13,37,39].

### 1.6. MenSC-Based Therapy for Vaginal Atrophy

In regenerative medicine, MSC are the most frequently used cell type due to their remarkable characteristics, including multilineage differentiation and paracrine effect capabilities [13,38]. This population of stem cells can be found in many tissues such as bone marrow, umbilical cord, periodontal ligament, and adipose tissue (adipose-derived mesenchymal stem cells, ADMSC). In recent years, both autologous and allogeneic ADMSC have been approved for the treatment of several chronic degenerative, inflammatory, and age-related diseases. ADMSC are very attractive considering their suitable direct use after isolation and easy accessibility to their source—body fat. In addition, autologous mechanically fragmented adipose tissue (MFAT) has great secretome activity in growth factors and cytokines, predominantly through small adipose clusters with intact vessels and perivascular niche [50].

Previous studies have reported that therapies based on stem cells, especially ADMSC, have positive outcomes regarding symptoms and signs associated with vaginal atrophy by promoting angiogenesis, vaginal restoration, and the proliferation of vaginal mucosa cells [44,50,51,52].

The first large long-term (36 months) study performed by Casarotti et al. evaluated the effectiveness and safety of a single subcutaneous vulvar injection of MFAT using the Lipogems^®^ medical device, which harvests, processes, and transfers adipose tissue, in 35 post-menopausal women with disorders related to genitourinary atrophy. After 9–12 months following MFAT injection, 90% of patients recovered completely from all symptoms and no relapse was reported in up to the third year of follow-up [11].

Recently, in a cohort of 11 post-menopausal patients who received one session of multiple injections of mechanically fragmented adipose tissue using the SEFFIGYN™ device, Mantovani et al. observed a notable attenuation of symptoms related to vaginal atrophy and an improvement in vulvar trophism in all patients during a 5-month follow-up period [38].

Zhang Y. et al. subjected 12 rhesus macaques to bilateral ovariectomy to induce menopause-related vaginal injury. Three months after surgery, in situ injection of umbilical cord-derived MSC was performed to evaluate changes of extracellular matrix, microvascular density, and smooth muscle in the vaginal tissue. Their results showed that MSC injection significantly increases extracellular matrix fibers, particularly collagen I and elastin, as well as microvascular density [12,50].

In 2019, Kasap et al. reported that ADMSC and bone marrow-derived MSC were able to improve vaginal mucosal epithelial thickness in a menopause rat model. Furthermore, VEGFR-1, VEGF, Bax, ER-α, and Bcl-2 staining levels were increased after ADMSC injection, both in vaginal epithelium and connective tissue [50].

To study the structure of vaginal mucosal atrophy, Faruk et al. injected intravaginal MSC in 55 induced ovariectomized rats, showing an increase in the mean thickness and regeneration of vaginal epithelium [53].

Likewise, Onesti et al. evaluated the efficacy of ADMSC-based therapy in eight patients with vulvar dystrophy. Significant histological vulvar trophism enhancement was observed in all patients, who also reported symptoms improvement associated with dystrophic areas throughout the 2-year follow-up [52].

The previously described exclusive properties of MenSC, including extensive expansion and differentiation capacities, high proliferative rate, genomic stability, low immunogenicity, paracrine effects, and anti-inflammatory and immunomodulatory functions with the ability to migrate into the injury site, have placed MenSC as a potent therapeutic candidate (Figure 3). Moreover, the easy and regular acquisition by non-invasive techniques and the lack of ethical issues compared to other source of MSC have attracted researchers’ attention for the use of this stem cell population in cellular therapies.

There are fundamental aspects that need to be taken into account when MenSC-based vulvovaginal atrophy therapy is considered for BRCA mutation carriers prior to RRSO. To obtain MenSC, the menstrual blood must be collected [54]. Thus, this novel treatment can exclusively be feasible if MenSC are obtained prior to RRSO. For this reason, one of the challenges that this stem cell-based therapy faces is the transfer of MenSC from bench to bedside without compromising product potency and quality. The cryopreservation of these cells seems an attractive solution to this issue with proven positive results, even though discrepancies remain in the literature regarding the efficacy of fresh versus cryo-MenSC [54]. Cryopreservation of MenSC can be achieved by controlled-rate or uncontrolled freezing techniques that cool samples to a temperature of −80 °C prior to ultra-low temperature storage transfer. This preservation procedure has the advantage of offering a longer shelf life and storage of great amounts of MenSC, allowing this autologous therapy in women who stop menstruating, after RRSO, or even after spontaneous menopause [54]. However, the optimal cryopreservation protocol is still under development and the long-term viability of cryo-MenSC should be carefully evaluated.

To date, there are no more than 10 clinical trials registered exploring the MenSC role in regenerative medicine, but none is related to the potential use of these stem cells in vaginal atrophy [55].

In summary, the theory behind the clinical efficacy of MenSC-based therapy for vaginal atrophy is based on the assumptions summarized in Table 1.

Not enough research has been conducted on the adverse reactions of MenSC-based therapy, and whether malignant transformation will occur after autologous transplantation. Nonetheless, a meta-analysis based on 62 prospective studies about the adverse events after MSC administration showed no association between this therapy and cancer or mortality incidence, and this analysis reported transient fever as the most common major side effect [55,56]. For this reason, robust evidence on this issue is mandatory in order to consider MenSC as a suitable therapy weapon for vaginal atrophy in patients with a predisposition to malignant disease, such as BRCA mutation carriers.

Regarding patient safety, menstrual blood samples collection, MenSC isolation, and transplantation must be performed under aseptic conditions according to good manufacturing practice standards.

Taking into account the existence of MenSC heterogeneity related to donor variability, diverse processes of cell culture, and environmental settings (including personal operation, injection method, epidemiological background, therapy duration, age, hormonal status, and overall health status), the management of clinical research concerning MenSC-based therapy should become normative and strict. Therefore, it is extremely important to further perform robust clinical trials to evaluate MenSC high-quality molecular markers, the optimal dose of MenSC, patterns of injection, and appropriate times for checkpoint and long-term adverse outcomes monitoring, so that the standardization and validation of these cells in vaginal atrophy therapy occurs, allowing a novel therapeutic weapon to arise.

## 2. Conclusions

The majority of BRCA1 and BRCA2 mutation carriers are subjected to risk-reducing bilateral salpingo-oophorectomy in premenopausal ages. The subsequent surgical menopause drastically leads to the onset of menopausal complaints, where vaginal atrophy-related symptoms play a major role. Due to the breast cancer risk profile of these patients, hormonal therapy is avoided in most cases. To date, the non-hormonal available therapies to treat vulvovaginal atrophy-related symptoms and improve the quality of life of these patients do not have sufficient efficacy to be considered an optimal solution.

MenSC-based therapy for vaginal atrophy has become a research hotspot, with promising results in this field. The unique properties of these cells, including their potential to be cryopreserved for an extended period of time for future use, high proliferative rates, low immunogenicity, and periodic acquisition in a non-invasive manner, have attracted researchers’ attention. On the other hand, the in vivo results that were comprehensively exposed in this review clearly highlight the need for further investigation into the therapeutical potential of MenSC for vaginal atrophy, in order to safely implement this novel therapeutic weapon in the market, especially when it concerns eventual malignant transformation.

## Figures and Tables

**Figure 1 ijms-25-01025-f001:**
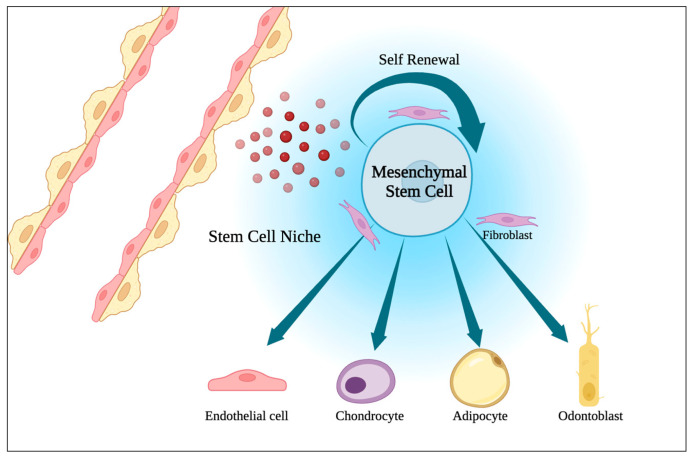
Representation of a stem cell niche (the place where humoral, neuronal, paracrine, physical, and metabolic pathways interact to regulate stem cell fate) and multipotency of mesenchymal stem cells.

**Figure 2 ijms-25-01025-f002:**
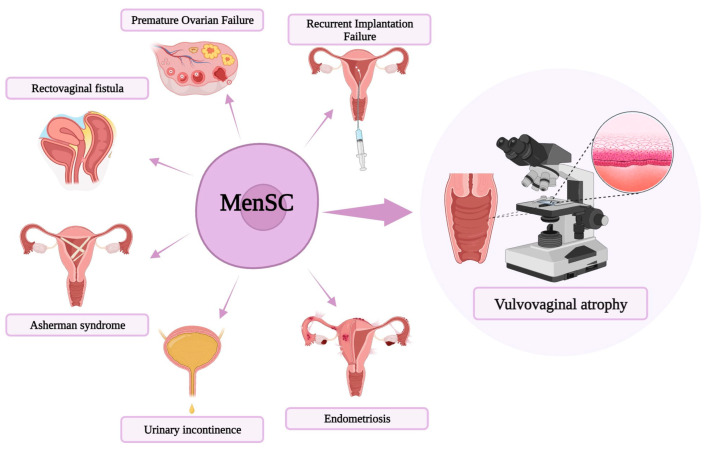
Schematic diagram of menstrual blood-derived stem cells (MenSC)-based treatments in a variety of gynecologic diseases, including premature ovarian failure, repeated implantation failure, vulvovaginal atrophy, endometriosis, urinary incontinence, Asherman syndrome, and rectovaginal fistula.

**Figure 3 ijms-25-01025-f003:**
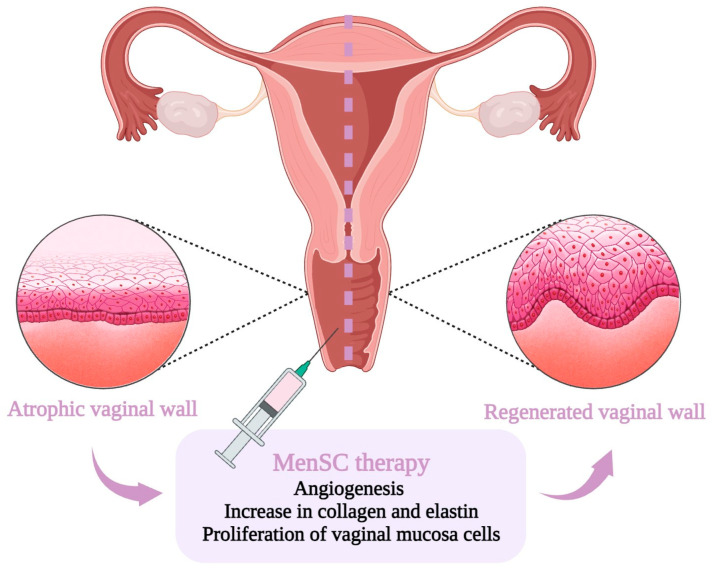
Schematic representation of the potential therapeutic strategies utilizing menstrual blood-derived stem cells (MenSC)-based treatment for vulvovaginal atrophy.

**Table 1 ijms-25-01025-t001:** Menstrual blood-derived stem cells properties beneficial for vaginal atrophy treatment.

MenSC ^1^ Property	Therapeutical Assumption
Regenerative	Help repair and regenerate the vaginal lining, increasing its thickness and reducing dryness.
Anti-inflammatory	Minimize the discomfort and pain associated with vaginal atrophy.
Pro-angiogenic	Improve the overall health of the vaginal tissue, leading to increased lubrication and elasticity.
Collagen production	Enhance the strength and elasticity of the vaginal walls.

^1^ MenSC, Menstrual blood-derived stem cell.

## Data Availability

Data are contained within the article.

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
