# Peer review of "The Concept behind the Suitability of Menstrual Blood-Derived Stem Cells for the Management of Vaginal Atrophy among BRCA Mutation Carriers after RRSO"

_ijms, 2024, doi:10.3390/ijms25021025_

Round 1

Reviewer 1 Report

Comments and Suggestions for Authors

As Menstrual SC-based therapy is not used yet for vaginal atrophy, it is not understanding the main idea of  article.

To persuade a reability of MenSC for therapy of vaginal disorders  It will be necessary to describe results of data regarding necessary characteristics of cells in model experiments in vitro and/or in vivo.

What will be the dose and protocol of MenSC injection for therapy of  vaginal atrophy ? Will it be  local, regional or even systemic therapy ?

Does MenSC require any special priming to increase its regenerative  capacity for vagina tissues regeneration?

Author Response

Dear reviewer,

We would like to thank you for your time and kind review. All your constructive comments and suggestions were considered and addressed.

As Menstrual SC-based therapy is not used yet for vaginal atrophy, it is not understanding the main idea of  article.

To persuade a reability of MenSC for therapy of vaginal disorders  It will be necessary to describe results of data regarding necessary characteristics of cells in model experiments in vitro and/or in vivo.

Author’s response: Thank you very much for your suggestion. According to Ru-Lin Huang et. al (1) we can conclude that MenSC apart from having a “notable capacity of enhanced angiogenesis and induction of tissue fibrosis, exert immunomodulatory activities and anti-inflammatory effects primarily through paracrine crosstalk with innate and adaptive immune cells.”. Additionally, Luz-Crawford et. al clearly state that MenSCs possess a low immunogenicity profile, as these cells express low levels of HLA-ABC and do not express HLA-DR in vivo (2). Moreover, Bozorgmehr et. al show that, in vitro, MenSCs inhibit differentiation and maturation of dendritic cells from human monocytes through secretion of IL-6 and IL-10 (3). All of the abovementioned authors have detailed the methodology  used in their research regarding the laboratory protocol for MenSC manipulation in the experiments. We decided not to include this detailed information in our review as it was not our intention to extensively describe the laboratory work behind the use of MenSC in regenerative medicine. Thank you for your optimizing our work with.

What will be the dose and protocol of MenSC injection for therapy of  vaginal atrophy ? Will it be  local, regional or even systemic therapy ?

Author’s response: We would like to thank you for your insightful comment. Therapy with MenSC should be performed by local injection of MenSC suspension in vaginal submucosal tissue, using a syringe with microneedles of 28G, in 4 to 6 longitudinal surface axes of vaginal surface, spaced 1-2 cm each.

Does MenSC require any special priming to increase its regenerative capacity for vagina tissues regeneration?

Author’s response: Once again, thanks for your comment. No special priming is required. Nevertheless cell priming to improve MenSC function with LL-37 and bioactive lipid shingosine-1-phosphate (S1P) demonstrates enhanced angiogenesis by the secretion of VEGFα, CXCR4, PDGF, HGF, and angiopoietin-1 Curcumin Adipose tissue in mouse models. So, previous MenSC conditioning could be useful in order to improve regenerative effects in vagina tissue repairing.

References:

  • Huang RL, Li Q, Ma JX, Atala A, Zhang Y. Body fluid-derived stem cells - an untapped stem cell source in genitourinary regeneration. Nat Rev Urol. 2023 Dec;20(12):739-761. doi: 10.1038/s41585-023-00787-2. Epub 2023 Jul 6. PMID: 37414959.
  • Luz-Crawford P, Torres MJ, Noël D, Fernandez A, Toupet K, Alcayaga-Miranda F, Tejedor G, Jorgensen C, Illanes SE, Figueroa FE, Djouad F, Khoury M. The immunosuppressive signature of menstrual blood mesenchymal stem cells entails opposite effects on experimental arthritis and graft versus host diseases. Stem Cells. 2016 Feb;34(2):456-69. doi: 10.1002/stem.2244. Epub 2015 Dec 3. PMID: 26528946.
  • Bozorgmehr, M. et al. Menstrual blood-derived stromal stem cells inhibit optimal generation and maturation of human monocyte-derived dendritic cells. Immunol. Lett. 162, 239–246 (2014)

Reviewer 2 Report

Comments and Suggestions for Authors

Reviewer statement:

Menstrual blood-derived stem cells are suitable to manage vaginal atrophy among BRCA mutation carriers after RRSO

Women carrying Breast Cancer gene 1 (BRCA1) and 2 (BRCA2) mutations have an increased risk for developing hereditary breast and ovarian cancer. Bilateral removal of the ovaries and fallopian tubes also known as bilateral salpingo-oophorectomy is a surgical procedure that is recommended and performed in women carrying with BRCA1 and BRCA2 mutations. Risk-reducing bilateral salpingo-oophorectomy (RRSO) is the most efficient strategy for ovarian and fallopian tube cancers prevention in  BRCA mutation carriers. Consequences of RSSO include detrimental menopausal consequences, such as increased risk of heart disease, osteoporosis, vasomotor symptoms, sexual dysfunction and neurocognitive deterioration. In BRCA mutation carriers hormonal therapies, including  hormone replacement therapy are generally avoided, even though the evidence for avoidance is limited. The authors report a review aiming to update the current knowledge and research regarding the potential value of menstrual blood-derived stem cells in the therapy of vaginal atrophy, namely among BRCA mutation carriers submitted to RRSO,  which is interesting, important and clinical relevant. This review can add valuable information to the readers, updating the available knowledge  and research on this topic.

Title: The title does not completely reflects the study reported; the title is too strong seeming to draw firm conclusion.

1.      After reading the article as a reader,  I cannot conclude that Menstrual blood-derived stem cells are suitable to manage vaginal atrophy. The authors should adjust the title and preferably report the type of study conducted in the title.

Overall: The paper is written in clear and concise English making the article attractive to read. The authors should be complimented for doing this.

Abstract : see overall remarks and remarks throughout the article

Introduction:

The introduction section is attractive to read form a reader point of view, explaining the background and reason for conducting this study. Despite , there is one important points from a readers point of view.

1.      The authors report in line  the following: “ Due to the inherent breast cancer risk  predisposition among BRCA mutations carriers, estrogen-based therapies are generally  avoided, even though this issue remains controversial (ref).”  The authors should report the reference to this statement. Please do so.

2.      Furthermore, the sentence reported in line 57-59 also requires references. Please provide the necessary references.

Article:

This section is well written and was easy to read. Despite , there are some points needing explanation and or clarification from a readers point of view to be able to understand and for the interpretation of the presented results.

3.      The title : “ i. Treatment options for vaginal atrophy after RRSO” should be adjusted as it is the only one who starts with i.  

4.      The authors report the following in line 195-197: “ Furthermore, patients may find the breast cancer risk profile of topical estrogen more acceptable in comparison to that of systemic combined HRT (19). “ The systemic exposure of topical estrogen is essential and should be mentioned by the authors. Is there evidence for systemic exposure of topical estrogen?

5.      The authors report the following in line 408-409 : “ Not enough research has been done on the adverse reactions of MenSC-based therapy, and whether malignant transformation will occur after authologous transplantation.” This is the reason to be cautious with firm conclusions on the use of MenSC-based therapy as there are so much uncertainty. Please elucidate thoroughly on this point in the whole article, especially in the discussion section.

Conclusion

No comments.

Tables and Figure

No comments.

Author Response

Dear reviewer,

We would like to thank you for your time and kind review. All your constructive comments and suggestions were considered and addressed.

Title: The title does not completely reflects the study reported; the title is too strong seeming to draw firm conclusion.

  1. After reading the article as a reader,  I cannot conclude that Menstrual blood-derived stem cells are suitable to manage vaginal atrophy. The authors should adjust the title and preferably report the type of study conducted in the title.

Author’s response: We would like to thank you for your insightful contribution to adjust the title of the article. In this way, we changed it to “The concept behind the suitability of menstrual blood-derived stem cells for the management of vaginal atrophy among BRCA mutation carriers after RRSO”

Overall: The paper is written in clear and concise English making the article attractive to read. The authors should be complimented for doing this.

Abstract : See overall remarks and remarks throughout the article

Introduction: The introduction section is attractive to read form a reader point of view, explaining the background and reason for conducting this study. Despite , there is one important points from a readers point of view.

  1. The authors report in line  the following: “ Due to the inherent breast cancer risk  predisposition among BRCA mutations carriers, estrogen-based therapies are generally  avoided, even though this issue remains controversial (ref).”  The authors should report the reference to this statement. Please do so.
  2. Furthermore, the sentence reported in line 57-59 also requires references. Please provide the necessary references.

Author’s response to both points 1. and 2.: Once again, we would like to thank you for your careful and kind revision of our article. It was a lapse of us not entering the references in the lines you mentioned. We have made the proper changes in the manuscript.

Article: This section is well written and was easy to read. Despite, there are some points needing explanation and or clarification from a readers point of view to be able to understand and for the interpretation of the presented results.

  1. The title : “ i. Treatment options for vaginal atrophy after RRSO” should be adjusted as it is the only one who starts with i.  

Author’s response: We completely agree with you and have deleted the ‘i.’ from the title.

  1. The authors report the following in line 195-197: “ Furthermore, patients may find the breast cancer risk profile of topical estrogen more acceptable in comparison to that of systemic combined HRT (19). “ The systemic exposure of topical estrogen is essential and should be mentioned by the authors. Is there evidence for systemic exposure of topical estrogen?

Author’s response: In order to address your insightful suggestion, we mention the data from a recent systematic review and meta-analysis performed by Comini et. al on this topic: “Safety and Serum Estradiol Levels in Hormonal Treatments for Vulvovaginal Atrophy in Breast Cancer Survivors: A Systematic Review and Meta-Analysis”.

  1. The authors report the following in line 408-409 : “ Not enough research has been done on the adverse reactions of MenSC-based therapy, and whether malignant transformation will occur after authologous transplantation.” This is the reason to be cautious with firm conclusions on the use of MenSC-based therapy as there are so much uncertainty. Please elucidate thoroughly on this point in the whole article, especially in the discussion section.

Author’s response: We carefully reviewed the sentences with firm conclusions on the use of MenSC-based therapy throughout the text and made the appropriate changes, especially in the discussion section as kindly recommended.

Conclusion: No comments.

Tables and Figure: No comments.

Round 2

Reviewer 1 Report

Comments and Suggestions for Authors

Some additional information was added to increase power of discussion the problem. It sounds reasonable and adequate.